# Convex Approximation of Two-Layer ReLU Networks for Hidden State Differential Privacy

**Rob Romijnders**
University of Amsterdam

**Antti Koskela**
Nokia Bell Labs

## Abstract

The hidden state threat model of differential privacy (DP) assumes that the adversary has access only to the final trained machine learning (ML) model, without seeing intermediate states during training. However, the current privacy analyses under this model are restricted to convex optimization problems, reducing their applicability to multi-layer neural networks, which are essential in modern deep learning applications. Notably, the most successful applications of the hidden state privacy analyses in classification tasks have only been for logistic regression models. We demonstrate that it is possible to privately train convex problems with privacy-utility trade-offs comparable to those of 2-layer ReLU networks trained with DP stochastic gradient descent (DP-SGD). This is achieved through a stochastic approximation of a dual formulation of the ReLU minimization problem, resulting in a strongly convex problem. This enables the use of existing hidden state privacy analyses and provides accurate privacy bounds also for the noisy cyclic mini-batch gradient descent (NoisyCGD) method with fixed disjoint mini-batches. Empirical results on benchmark classification tasks demonstrate that NoisyCGD can achieve privacy-utility trade-offs on par with DP-SGD applied to 2-layer ReLU networks.

## 1 Introduction

In differentially private (DP) machine learning (ML), the DP-SGD algorithm (see e.g., Abadi et al., 2016) has become a ubiquitous tool to obtain ML models with strong privacy guarantees. The guarantees for DP-SGD are obtained by clipping gradients and by adding normally distributed noise to the randomly sampled mini-batch of gradients. The parameters for the DP guarantee are then found by a composition analysis of the iterative algorithm (see, e.g., Zhu et al., 2022).

One weak point of the composition analysis of DP-SGD is the assumption that the adversary has access to all the intermediate results of the training iteration. This assumption is unnecessarily strict as in many practical scenarios only the final model needs to be revealed (Andrew et al., 2024). Another weakness is that DP-SGD requires either full batch training or random subsampling. Both are computationally not favourable, as full batch usually does not fit in memory and random subsampling implies an unbalanced use of compute resources and increased implementation complexity (Chua et al., 2024a). In contrast, for noisy cyclic gradient descent (NoisyGD) with disjoint mini-batches, the compute is balanced between mini-batches and can be implemented efficiently. Having high privacy-utility ML models trained with NoisyCGD would give an alternative for DP-SGD that is more practical and efficient to implement (Chua et al., 2024a).

The so-called hidden state threat model of DP considers releasing only the final model of the training iteration. Existing $(\varepsilon, \delta)$-DP analyses in the literature are only applicable for convex problems such as logistic regression which is the highest performing model considered in the literature (see, e.g., Chourasia et al., 2021; Bok et al., 2024). When training models with DP-SGD, however, one quickly finds that the model performance of commonly used convex models is inferior compared to neural networks. A natural question arises whether convex approximations of minimization problems for

multi-layer neural networks can be made while preserving model performance under privacy. This work explores such an approximation for the two-layer ReLU neural network. We build on the findings of Pilanci and Ergen (2020), which demonstrate the existence of a convex dual formulation for the two-layer ReLU minimization problem when the hidden layer has sufficient width.

The privacy amplification by iteration analysis for convex private optimization, introduced by Feldman et al. (2018), provides privacy guarantees in the hidden state threat model. However, these analyses (Sordello et al., 2021; Asoodeh et al., 2020; Chourasia et al., 2021; Altschuler and Talwar, 2022) remain challenging to apply in practice, as they typically require a large number of training iterations to obtain tighter DP guarantees than those of DP-SGD. Chourasia et al. (2021) improve this analysis using Rényi DP for full-batch training with DP guarantees, while Ye and Shokri (2022) offer a similar analysis for shuffled mini-batch DP-SGD. Recently, Bok et al. (2024) provided an $f$-DP analysis for a class of algorithms, which we will leverage to analyze NoisyCGD.

Our main contributions are the following:

- By integrating two seemingly unrelated approaches, a convex reformulation of ReLU networks and the privacy amplification by iteration DP analysis, we show that it is possible to obtain similar privacy-utility trade-offs in the hidden state threat model of DP as by applying DP-SGD to two-layer ReLU networks and using well-known composition results.

- We provide approximations for the convex reformulation to facilitate DP analysis and show that the resulting strongly convex model has the required properties for hidden state analysis.

- We give the first high-privacy-utility trade-off results for image classification tasks using a hidden state DP analysis. In particular, we present results for NoisyCGD with disjoint mini-batches. This allows for more practical applications of DP ML.

- We run a theoretical utility analysis of gradient descent with DP guarantees when applied to the convex approximation, in the context of the random data model.

## 1.1 Further Related Literature

Differentially private machine learning has been extensively studied in recent years. The most widely used approach is DP-SGD (Abadi et al., 2016), which clips and adds noise to gradients during training. Several works have analyzed and improved DP-SGD through techniques such as adaptive clipping (Andrew et al., 2021), better composition analyses (Dong et al., 2022; Koskela et al., 2020; Zhu et al., 2022; Gopi et al., 2021), or privacy amplification by iteration (Feldman et al., 2018). For convex problems, alternative approaches include sufficient statistics perturbation (Wang, 2018; Amin et al., 2023), objective perturbation (Chaudhuri et al., 2011), and output perturbation (Wu et al., 2017). Most prior work focuses on the standard DP definition rather than hidden state DP. Another alternative to DP-SGD is the Differentially Private Follow-the-Regularized-Leader (Kairouz et al., 2021), but it comes with significant additional memory and compute cost due to sampling of correlated noise from a matrix mechanism, and potentially significant communication overhead in the federated learning setting. A recent line of work considers DP guarantees for DP-SGD with disjoint batches. Chua et al. (2025) and Choquette-Choo et al. (2025) use Monte Carlo methods to estimate privacy guarantees under the so-called balls-and-bins sampling, where Poisson sampling assigns data points to disjoint batches. However, these simulations are computationally intensive and do not lead to provable upper bounds on DP parameters. Feldman and Shenfeld (2025) provide provable bounds for a generalization of this scheme, though their method also results in randomly sized batches. Thus, existing analyses do not accommodate fixed-size batches and, in particular, unshuffled data, further motivating our approach, which yields high-utility convex models for which NoisyCGD can be analyzed accurately.

Our work builds on recent advances in convex approximations of neural networks. The connection between two-layer ReLU networks and convex optimization was established by Bengio et al. (2005); Bach (2017) and further developed by Pilanci and Ergen (2020). Several works propose methods to train neural networks through convex optimization (Ergen and Pilanci, 2020; Ergen et al., 2023). However, the privacy implications of these convex formulations have not been thoroughly explored before our work. The closest related work is (Kim and Pilanci, 2024), which analyzes connections between stochastic dual forms and ReLU networks but does not consider privacy.

## 2   Preliminaries

We denote a dataset containing $n$ data points as $D = (z_1, \ldots, z_n)$. We say $D$ and $D'$ are neighboring datasets if they differ in exactly one element (denoted as $D \sim D'$). A mechanism $\mathcal{M} : \mathcal{X} \to \mathcal{O}$ is $(\varepsilon, \delta)$-DP if the output distributions for neighboring datasets are always $(\varepsilon, \delta)$-indistinguishable (Dwork et al., 2006).

**Definition 2.1.** Let $\varepsilon \geq 0$ and $\delta \in [0, 1]$. Mechanism $\mathcal{M} : \mathcal{X} \to \mathcal{O}$ is $(\varepsilon, \delta)$-DP if for every pair of neighboring datasets $D \sim D'$ and for every measurable set $E \subset \mathcal{O}$,

$$\mathbb{P}(\mathcal{M}(D) \in E) \leq \mathrm{e}^\varepsilon \mathbb{P}(\mathcal{M}(D') \in E) + \delta.$$

We call $\mathcal{M}$ tightly $(\varepsilon, \delta)$-DP, if there does not exist $\delta' < \delta$ such that $\mathcal{M}$ is $(\varepsilon, \delta')$-DP.

The DP guarantees can alternatively be described using the hockey-stick divergence. For $\alpha > 0$ the hockey-stick divergence $H_\alpha$ from a distribution $P$ to a distribution $Q$ is defined as $H_\alpha(P\|Q) = \int \max\{P(t) - \alpha \cdot Q(t), 0\} \, \mathrm{d}t$. The $(\varepsilon, \delta)$-DP guarantee, in Def. 2.1, can be characterized using the hockey-stick divergence: if we can bound the divergence $H_{\mathrm{e}^\varepsilon}(\mathcal{M}(D)\|\mathcal{M}(D'))$ accurately, we also obtain accurate $\delta(\varepsilon)$-bounds. We also refer to $\delta_\mathcal{M}(\varepsilon) := \max_{D \sim D'} H_{\mathrm{e}^\epsilon}(\mathcal{M}(D)\|\mathcal{M}(D'))$ as the *privacy profile* of mechanism $\mathcal{M}$. To accurately bound the hockey-stick divergence of compositions, we need to so-called dominating pairs of distributions.

**Definition 2.2** (Zhu et al. 2022)**.** A pair of distributions $(P, Q)$ is a *dominating pair* of distributions for mechanism $\mathcal{M}(D)$ if for all neighboring datasets $D$ and $D'$ and for all $\alpha > 0$,

$$H_\alpha(\mathcal{M}(D)\|\mathcal{M}(D')) \leq H_\alpha(P\|Q).$$

If the equality holds for all $\alpha$ for some $D \sim D'$, then $(P, Q)$ is a tightly dominating pair of distributions. We get upper bounds for DP-SGD compositions using the dominating pairs of distributions using the following composition result.

**Theorem 2.3** (Zhu et al. 2022)**.** *If $(P, Q)$ dominates $\mathcal{M}$ and $(P', Q')$ dominates $\mathcal{M}'$, then $(P \times P', Q \times Q')$ dominates the adaptive composition $\mathcal{M} \circ \mathcal{M}'$.*

To convert the hockey-stick divergence from $P \times P'$ to $Q \times Q'$ into an efficiently computable form, we consider so called privacy loss random variables (PRVs) and use Fast Fourier Technique-based methods (Koskela et al., 2021; Gopi et al., 2021) to numerically evaluate the convolutions appearing when summing the PRVs and evaluating $\delta(\varepsilon)$ for the compositions.

**Gaussian Differential Privacy.**   For the privacy accounting of the noisy cyclic mini-batch GD, we use the bounds by Bok et al. (2024) that are stated using the Gaussian differential privacy (GDP). Informally speaking, a mechanism $\mathcal{M}$ is $\mu$-GDP, $\mu \geq 0$, if for all neighboring datasets the outcomes of $\mathcal{M}$ are not more distinguishable than two unit-variance Gaussians $\mu$ apart from each other (Dong et al., 2022). We consider the following formal characterization of GDP.

**Lemma 2.4** (Dong et al. 2022, Cor. 2.13)**.** *A mechanism $\mathcal{M}$ is $\mu$-GDP if and only it is $(\varepsilon, \delta)$-DP for all $\varepsilon \geq 0$, where*

$$\delta(\varepsilon) = \Phi\left(-\frac{\varepsilon}{\mu} + \frac{\mu}{2}\right) - \mathrm{e}^\varepsilon \Phi\left(-\frac{\varepsilon}{\mu} - \frac{\mu}{2}\right).$$

### 2.1   DP-SGD with Poisson Subsampling

One iteration of DP-SGD with Poisson subsampling is given by

$$\theta_{j+1} = \theta_j - \eta_j \cdot \left(\frac{1}{b} \sum\nolimits_{x \in B_j} \mathrm{clip}(\nabla\mathcal{L}(x, \theta_j), C) + Z_j\right),$$

where $C > 0$ denotes the clipping constant, $\mathrm{clip}(\cdot, C)$ the clipping function that clips gradients to have the 2-norm at most $C$, $\mathcal{L}$ the loss function, $\theta$ the model parameters, $\eta_j$ the learning rate at iteration $j$, $B_j$ the mini-batch at iteration $j$ sampled with Poisson subsampling with the subsampling ratio $b/n$, $b$ the expected size of each mini-batch and $Z_j \sim \mathcal{N}(0, \frac{C^2\sigma^2}{b^2}I_d)$ the noise vector.

We adopt the substitute neighborhood relation and apply the result of Lebeda et al. (2024):

**Lemma 2.5** (Lebeda et al. 2024). *Suppose a pair of distributions $(P, Q)$ is a dominating pair of distributions for a mechanism $\mathcal{M}$ and denote the Poisson subsampled mechanism $\widetilde{\mathcal{M}} := \mathcal{M} \circ S^q_{Poisson}$, where $S^q_{Poisson}$ denotes the Poisson subsampling with subsampling ratio $q$. Then, under the $\sim$-neighbouring relation, the pair of distributions $(P, Q)$, where $P = (1-q) \cdot \mathcal{N}(0, \sigma^2) + q \cdot \mathcal{N}(1, \sigma^2)$ and $Q = (1-q) \cdot \mathcal{N}(0, \sigma^2) + q \cdot \mathcal{N}(-1, \sigma^2)$ is a dominating pair of distributions for $\widetilde{\mathcal{M}}$.*

Combined with Lemma 2.3 and numerical accountants, this yields tight $(\varepsilon, \delta)$ bounds for DP-SGD with Poisson subsampling under the substitute neighborhood relation.

## 2.2 Guarantees for the Final Model and Noisy Cyclic Mini-Batch GD

We next consider privacy amplification by iteration (Feldman et al., 2018), which gives privacy guarantees for the final model of the training iteration. Recent results by Bok et al. (2024) are applicable to the noisy cyclic mini-batch gradient descent (NoisyCGD) for which one epoch of training is described by the iteration:

$$\theta_{j+1} = \theta_j - \eta \left( \frac{1}{b} \sum_{x \in B_j} \nabla_\theta f(\theta_j, x) + Z_j \right)$$

where $Z_j \sim \mathcal{N}(0, \sigma^2 I_d)$ and the data $D$ is divided into disjoint batches $B_1, \ldots, B_k$, each of size $b$. The analysis by Bok et al. (2024) also considers the substitute neighborhood relation of datasets. A central element for the DP analysis is the gradient sensitivity:

**Definition 2.6.** A loss function family $\mathcal{F}$ has a gradient sensitivity $L$ if $\sup_{f,g \in \mathcal{F}} \|\nabla f - \nabla g\| \leq L$.

For example, for a family of loss functions of the form $h_i + r$, where $h_i$'s are $L$-Lipschitz loss functions and $r$ is a regularization function, the sensitivity equals $2L$. We will use the following result to analyse the DP guarantees of NoisyCGD. Recall that a function $f$ is $\beta$-smooth if $\nabla f$ is $\beta$-Lipschitz, and it is $\lambda$-strongly convex if the function $g(x) = f(x) - \frac{\lambda}{2} \|x\|_2^2$ is convex.

**Theorem 2.7** (Bok et al. 2024, Thm. 4.5). *Consider $\lambda$-strongly convex, $\beta$-smooth loss functions with gradient sensitivity $L$. Then, for any $\eta \in (0, 2/\beta)$, NoisyCGD is $\mu$-GDP for*

$$\mu = \frac{L}{b\sigma} \sqrt{1 + c^{2k-2} \frac{1 - c^2}{(1 - c^k)^2} \frac{1 - c^{k(E-1)}}{1 + c^{k(E-1)}}},$$

*where $k = n/b$, $c = \max\{|1 - \eta\lambda|, |1 - \eta\beta|\}$ and $E$ denotes the number of epochs.*

We could alternatively use the RDP analysis by Ye and Shokri (2022). However, as also illustrated by the experiments of Bok et al. (2024), the bounds given by Thm. 2.7 lead to slightly lower $(\varepsilon, \delta)$-DP bounds for NoisyCGD. To benefit from the privacy analysis of Thm. 2.7 for NoisyCGD, we add an $L_2$-regularization term with a coefficient $\frac{\lambda}{2}$. This makes the loss function $\lambda$-strongly convex.

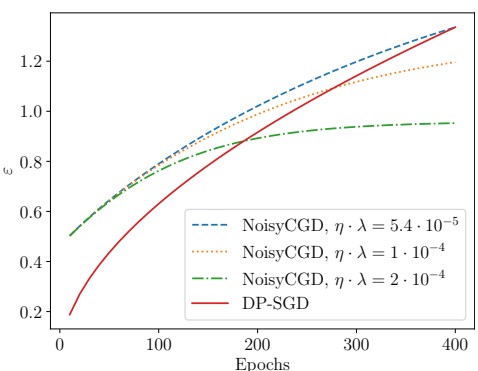

Finding suitable hyperparameter values for the learning rate $\eta$ and regularization parameter $\lambda$ is complicated by two aspects. The larger the regularization parameter $\lambda$ and the learning rate $\eta$ are, the faster the model 'forgets' the past updates and the quicker the $\varepsilon$-values converge. This is reflected in the GDP bound of Thm. 2.7 in the constant $c$, which generally equals $|1 - \eta\lambda|$. To benefit from the bounds of Thm. 2.7, the product $\eta\lambda$ should not be too small. Alternatively, when $\eta\lambda$ is too large, the 'forgetting' affects the model performance. We experimentally observe that the plateauing of the model accuracy and privacy guarantees occurs approximately simultaneously.

Figure 1: Values of the product of the learning rate $\eta$ and the $L_2$-regularization constant $\lambda$ that lead to tighter privacy bounds for the final model using Thm. 2.7, compared to the whole sequence of updates using the DP-SGD analysis. Here $n = 6 \cdot 10^4$, $b = 1000$, $\sigma = 15.0$ and $\delta = 10^{-5}$.

Figure 1 illustrates the privacy guarantees of NoisyCGD for a range of values for the product $\eta\lambda$. The $(\varepsilon, \delta)$-DP guarantees given by

Thm. 2.7 become smaller than those given by the Poisson subsampled DP-SGD with an equal batch size $b = 1000$ when $\sigma = 15.0$ and training for 400 epochs. For a given learning rate $\eta$, we can always adjust the value of $\lambda$ to have desirable $(\varepsilon, \delta)$-DP guarantees. To put the values of Fig. 1 into perspective, in experiments we observe that $\eta\lambda = 2 \cdot 10^{-4}$ is experimentally found to affect the model performance already considerably whereas $\eta\lambda = 1 \cdot 10^{-4}$ affects only weakly.

**Computational advantages of NoisyCGD.** NoisyCGD offers significant computational benefits over traditional DP-SGD. The standard DP-SGD requires Poisson sampling where each sample is included in a batch with an independent probability, leading to variable batch sizes. This variability creates practical challenges: For a target physical batch size of 256, one typically needs to set the expected batch size to 230 to handle size variations, resulting in an 11% throughput reduction. Additionally, approximately 4% of batches exceed the physical limit and must be processed separately as stragglers. This further reduces efficiency and increases the engineering complexity of experimental code.

While Chua et al. (2024b) proposed truncating oversized batches and accounting for this in the $\delta$ parameter of $(\varepsilon, \delta)$-DP, this approach also impacts throughput. For example, with a physical batch size of 256, $\varepsilon = 8$, $\delta = 10^{-6}$, and 10 training epochs, the expected batch size must be reduced to 166 according to their theorem. This constitutes a 35% loss of throughput. In contrast, NoisyCGD maintains constant batch sizes (except for possibly the final batch). This enables a simpler implementation in code and higher throughput while preserving privacy guarantees.

# 3 Convex Approximation of Two-Layer ReLU Networks

This section derives the strongly convex approximation of the 2-layer ReLU minimization problem and shows that the derived problem is amenable to privacy amplification by iteration analysis. Without loss of generality, we consider a 1-dimensional output network (e.g., a binary classifier), which can be extended to multivariate output networks later (see also Ergen et al., 2023).

## 3.1 Convex Duality of Two-layer ReLU Problem

Consider training a ReLU network (with hidden-width $m$) $f : \mathbb{R}^d \to \mathbb{R}$ (Pilanci and Ergen, 2020),

$$f(x) = \sum_{j=1}^{m} \phi(u_j^\top x)\alpha_j. \tag{3.1}$$

The weights are $u_i \in \mathbb{R}^d$, $i \in [m]$ and $\alpha \in \mathbb{R}^m$. The ReLU activation function is $\phi(t) = \max\{0, t\}$. For a vector $x$, $\phi$ is applied element-wise, i.e. $\phi(x)_i = \phi(x_i)$.

Suppose the dataset $D$ consists of $n$ tuples of the form $z_i = (x_i, y_i)$, $x_i \in \mathbb{R}^d$, $y_i \in \mathbb{R}$, for $i \in [n]$. Using the squared loss and $L_2$-regularization with a regularization constant $\lambda > 0$, the 2-layer ReLU minimization problem can be written as

$$\min_{\{u_i, \alpha_i\}_{i=1}^m} \frac{1}{2} \left\| \sum_{i=1}^{m} \phi(Xu_i)\alpha_i - y \right\|_2^2 + \frac{\lambda}{2} \sum_{i=1}^{m} (\|u_i\|_2^2 + \alpha_i^2), \tag{3.2}$$

where $X \in \mathbb{R}^{n \times d}$ denotes the matrix of the feature vectors, i.e., $X^\top = [x_1 \quad \ldots \quad x_n]$ and $y \in \mathbb{R}^n$ denotes the vector of labels.

The convex reformulation of the ReLU problem (3.2) is based on enumerating all the possible activation patterns of $\phi(Xu)$, $u \in \mathbb{R}^d$. The set of activation patterns that a ReLU output $\phi(Xu)$ can take for a data feature matrix $X \in \mathbb{R}^{n \times d}$ is described by the set of diagonal boolean matrices

$$\mathcal{D}_X = \{\Lambda = \operatorname{diag}(\mathbb{1}(Xu \geq 0)) : u \in \mathbb{R}^d\}, \tag{3.3}$$

where for $i \in [n]$, $\left(\mathbb{1}(Xu \geq 0)\right)_i = 1$, if $(Xu)_i \geq 0$ and 0 otherwise. The number of regions in a partition of $\mathbb{R}^d$ by hyperplanes that pass through the origin and are perpendicular to the rows of $X$ is $|\mathcal{D}_X|$. We have by Pilanci and Ergen (2020): $|\mathcal{D}_X| \leq 2r \left(\frac{e(n-1)}{r}\right)^r$, where $r = \operatorname{rank}(X)$.

Notice that $\mathcal{D}_X$ defined in Eq. (3.3) is the set of all possible boolean matrices, where each boolean matrix corresponds to a different pattern of ReLU activations for the features of the dataset (the rows of the matrix $X \in \mathbb{R}^{n \times d}$). Thus, the bound for $|\mathcal{D}_X|$ is a bound for the number of possible hyperplane arrangements, and thus it is only determined by the feature matrix $X$.

Let the number of activation patterns be $|\mathcal{D}_X| = M$, and denote $\mathcal{D}_X = \{\Lambda_1, \ldots, \Lambda_M\}$. Let $\lambda > 0$. The parameter space can then be partitioned into convex cones $\mathcal{V}_1, \ldots, \mathcal{V}_M$, where $\mathcal{V}_i = \{u \in \mathbb{R}^d : (2\Lambda_i - I)Xu \geq 0\}$. We consider the following convex optimization problem with group $\ell_2$–$\ell_1$ regularization:

$$\min_{v_i, w_i} \frac{1}{2} \left\| \sum_{i \in [M]} \Lambda_i X(v_i - w_i) - y \right\|_2^2 + \lambda \sum_{i \in [M]} (\|v_i\|_2 + \|w_i\|_2) \tag{3.4}$$

subject to $v_i, w_i \in \mathcal{V}_i$ for all $i \in [M]$, i.e., $(2\Lambda_i - I)Xv_i \geq 0$ and $(2\Lambda_i - I)Xw_i \geq 0$ for all $i \in [M]$.

Interestingly, for a sufficiently large hidden width $m$, the ReLU minimization problem (3.2) and the convex problem (3.4) attain the same minimal value (see Pilanci and Ergen, 2020, Thm. 1). Moreover, Pilanci and Ergen (2020) show that the optimal ReLU network weights can be recovered from the solution of the convex problem (3.4). Hence, for large enough $m$, the convex problem can be viewed as an equivalent convex reformulation of the ReLU problem (3.2), where $\{v_i\}_{i=1}^P$ and $\{w_i\}_{i=1}^P$ act as dual-like variables. Subsequent work, such as Mishkin et al. (2022), extends this equivalence to general convex loss functions $\mathcal{L}$. For simplicity, we restrict our discussion to the squared loss. We also note that similar convex formulations have been developed for two-layer convolutional networks (Bartan and Pilanci, 2019) and for multi-layer ReLU networks (Ergen and Pilanci, 2021).

## 3.2 Stochastic Approximation

Since $|\mathcal{D}_X|$ is generally an enormous number, stochastic approximations to the problem (3.4) have been considered (Pilanci and Ergen, 2020; Wang et al., 2022; Mishkin et al., 2022; Kim and Pilanci, 2024). Vectors $u_i \sim \mathcal{N}(0, I_d)$, $i \in [P]$, $P \ll M$, are then sampled randomly to construct the boolean diagonal matrices $\Lambda_1, \ldots, \Lambda_P$, $\Lambda_i = \mathrm{diag}(\mathbb{1}(Xu_i \geq 0))$, and the problem (3.4) is replaced by

$$\min_{v_i, w_i} \frac{1}{2} \left\| \sum_{i=1}^P \Lambda_i X(v_i - w_i), y \right\|_2^2 + \lambda \sum_{i=1}^P (\|v_i\|_2 + \|w_i\|_2) \tag{3.5}$$

such that for all $i \in [P]$: $v_i, w_i \in \mathcal{V}_i$, i.e., $(2\Lambda_i - I)Xw_i \geq 0$, and $(2\Lambda_i - I)Xv_i \geq 0$ for all $i \in [P]$. These constraints, however, are data-dependent, which makes private learning of the problem (3.5) difficult. Moreover, the overall loss function given by Eq. 3.5 is not generally strongly convex. This prevents using privacy amplification results such as Theorem 2.7 for NoisyCGD. We thus consider a strongly convex problem without any constraints and with a squared regularization term.

## 3.3 Stochastic Strongly Convex Approximation

Motivated by the needs of the DP analysis and the formulation given in (Wang et al., 2022), we consider global minimization of the loss function

$$\mathcal{L}(v, X, y) = \frac{1}{n} \sum_{j=1}^n \ell(v, x_j, y_j), \tag{3.6}$$

with

$$\ell(v, x_j, y_j) = \frac{1}{2} \left\| \sum_{i=1}^P (\Lambda_i)_{jj} x_j^\top v_i - y_j \right\|_2^2 + \frac{\lambda}{2} \sum_{i=1}^P \|v_i\|_2^2, \quad (\Lambda_i)_{jj} = \mathbb{1}(x_j^\top u_i \geq 0), \tag{3.7}$$

where $v = \{v_i\}_{i=1}^P$, $v_i \in \mathbb{R}^d$ for $i \in [P]$ denote the learnable parameters. We note that due to the convention of the literature, the regularization is added term-wise (Bok et al., 2024). The diagonal boolean matrices $\Lambda_1, \ldots, \Lambda_P \in \mathbb{R}^{n \times n}$ are again constructed by taking first $P$ i.i.d. samples $u_1, \ldots, u_P$, $u_i \sim \mathcal{N}(0, I_d)$, and then setting the diagonals of $\Lambda_i$'s: $(\Lambda_i)_{jj} = \max(0, \mathrm{sign}(x_j^\top u_i))$.

**The Model at Inference Time.** Having a sample $x \in \mathbb{R}^d$ at inference time, one uses the $P$ vectors $u_1, \ldots, u_P$ that were used for constructing the boolean diagonal matrices $\Lambda_i$, $i \in [P]$ during training. The prediction is carried out similarly using the function $g(x, v) = \sum_{i=1}^P \mathbb{1}(u_i^\top x \geq 0) \cdot x^\top v_i$.

**Practical Considerations.** In the experiments, we use the cross-entropy loss instead of the mean squared error for the loss function $\ell$. For $K$-dimensional outputs and labels, we employ $K$ independent linear models in parallel, each predicting one output dimension. The resulting model has parameter dimensionality $d \times P \times K$, where $d$ is the feature dimension and $P$ is the number of randomly chosen hyperplanes.

### 3.4 Meeting the Requirements of DP Analysis

In Eq. (3.7) each loss function $\ell(v, x_j, y_j)$, $j \in [n]$, depends only on the data entry $(x_j, y_j)$. By clipping the data sample-wise gradients $\nabla_v h(v, x_j, y_j)$, where $h(v, x_j, y_j) = \frac{1}{2} \left\| \sum_{i=1}^{P} (\Lambda_i)_{jj} x_j^\top v_i - y_j \right\|_2^2$, the loss function $\ell$ becomes $2L$-sensitive (see Def. 2.6). Appendix C shows that the loss function $\ell(v, x_j, y_j)$ is a loss function of a generalized linear model and thus we are allowed to use the analysis of Bok et al. (2024) when clipping the gradients since then the clipped gradients are gradients of another convex loss (Song et al., 2021). For the DP analysis, we must also analyze the loss function's convexity properties (3.7).

We have the following Lipschitz bound for the gradients.

**Lemma 3.1.** *The gradients of the loss function $\ell(v, x_j, y_j)$ given in Eq. (3.7) are $\beta$-Lipschitz continuous for $\beta = \|x_j\|_2^2 + \lambda$.*

Due to $L_2$-regularization, the loss function (3.7) is $\lambda$-strongly convex. The properties of $\lambda$-strong convexity and $\beta$-smoothness are preserved when clipping the sample-wise gradients $\nabla_v h(v, x_j, y_j)$ (Section E.2, Redberg et al., 2024). Therefore, the DP accounting Thm. 2.7 is applicable with the same convexity parameters when clipping the gradients $\nabla_v h(v, x_j, y_j)$.

## 4 Theoretical Analysis

From a theoretical standpoint, convex models are advantageous over non-convex ones in private optimization. State-of-the-art empirical risk minimization (ERM) bounds for private convex optimization are of the order $O(\frac{1}{\sqrt{n}} + \frac{\sqrt{p}}{\varepsilon n})$, where $n$ is the number of training data entries, $p$ the dimension of the parameter space and $\varepsilon$ the DP parameter (Bassily et al., 2019). In contrast, the bounds for non-convex optimization, which focus on finding stability points, are much worse, such as $O(\frac{1}{n^{1/3}} + \frac{p^{1/5}}{(\varepsilon n)^{2/5}})$ (Bassily et al., 2021) and $O\left(\frac{p^{1/3}}{(\varepsilon n)^{2/3}}\right)$ (Arora et al., 2023; Lowy et al., 2024).

We consider utility bounds with random data for the problem (3.7). The random data model is also commonly used in the analysis of private linear regression (see, e.g., Varshney et al., 2022). Leveraging classical DP-ERM results, we derive utility bounds that are tighter than those obtainable for 2-layer ReLU networks via non-convex optimization. In addition to the convexity benefits of DP-ERM, we utilize the approximability of our convex models: with enough random hyperplanes, the global minimum $\mathcal{L}(\theta^*, D)$ goes to zero with high probability.

### 4.1 Utility Bound for the Convex ReLU Approximation in the Random Data Model

Recently, Kim and Pilanci (2024) have given several results for convex problems under the assumption of random feature data, i.e., when $y \in \mathbb{R}^n$ is fixed and $X_{ij} \sim \mathcal{N}(0, 1)$ i.i.d. Based on their results, using $P = O\big((n \log n)/d\big)$ hyperplane arrangements, the stochastic objective in (3.7) attains a global minimum of zero with high probability. Consequently, standard DP-ERM bounds yield the result of Thm. 4.1. In future work, it will be interesting to see whether techniques from private linear regression (Liu et al., 2023; Avella-Medina et al., 2023; Varshney et al., 2022; Cai et al., 2021) could be used to eliminate the assumption of bounded gradients. Notice that Thm. 4.1 exhibits feature dimension-independence unlike existing DP-SGD bounds for the linear regression under the random data assumption that are $\widetilde{O}(\frac{d}{\sqrt{n}\varepsilon})$ (Thm. 1.2, Brown et al., 2024).

**Theorem 4.1.** *Consider the random data model where $y \in \mathbb{R}^n$ and the elements of the data matrix $X \in \mathbb{R}^{n \times d}$ are i.i.d. distributed as $X_{ij} \sim \mathcal{N}(0, 1)$ and consider applying the private gradient descent (repeated from literature in Alg. 1) to the strongly convex loss (3.6) with $P = O\big((n \log n)/d\big)$ hyperplane arrangements, and assume that the gradients are bounded by a constant $L > 0$. Let the ratio $c = \frac{n}{d} \geq 1$ be fixed. For any $\gamma > 0$, there exists $d_1$ such that for all $d \geq d_1$, with probability at least $1 - \gamma - \frac{1}{(2n)^8}$ (with $\widetilde{O}$ omitting the logarithmic factors)*

$$\mathcal{L}(\theta^{priv}, D) \leq \widetilde{O}\left(\frac{1}{\sqrt{n}\varepsilon}\right).$$

***Remark* 1.** *In Thm. 4.1, we assume that the ratio $c = n/d$ stays constant in order to be able to use the results of Kim and Pilanci (2024). One motivation for the assumption $c \geq 1$ comes from having a similar setting as Brown et al. (2024) since we make an explicit comparison to their utility bounds, where the dependency on the feature dimension $d$ appears. Brown et al. (2024) assume that $n = \Omega(d)$ (Thm. 1.2, Brown et al., 2024), which is implied by the assumption $n/d \geq 1$. The results are in this sense comparable, whereas with the assumption $n/d < 1$ there is no implication in any direction. As our Thm. 4.1 indicates, it is possible to get rid of the dependency on the parameter $d$ when training our proposed model with DP-SGD, and in this sense, our approach is better in the random data model than the private linear regression analyzed by Brown et al. (2024). Notice also that in our Thm. 4.1 there is no randomness assumption on the label vector $y \in \mathbb{R}^n$ unlike in (Thm. 1.2, Brown et al., 2024).*

*Another motivation for the assumption $n/d \geq 1$, or rather requirement, comes from the mathematical analysis: the current results by Kim and Pilanci (2024) that we use have to assume $n/d \geq 1$ for the model loss to have, with high probability, a unique zero global minimum for every possible label vector $y$ with a limited number of random hyperplanes $P = O(n \log n/d)$. If we assume $n/d < 1$, in a sense the problem becomes easier since the feature matrix $X$ becomes column rank deficient with high probability. Then, there also can be an infinite number of global minimizers. Moreover, the DP-SGD convergence results by Bassily et al. (2014) and Talwar et al. (2014), which we use in our analysis, do not require a unique global minimizer. It is an interesting question for future work how to include the case $n/d < 1$ in the utility analysis.*

## 5  Experimental Results

We compare the methods on standard image classification benchmarks: MNIST (LeCun et al., 1998), FashionMNIST (Xiao et al., 2017), and CIFAR10 (Krizhevsky et al., 2009). The MNIST dataset has a training dataset of $6 \cdot 10^4$ samples and a test dataset of $10^4$ samples. CIFAR10 has a training dataset size of $5 \cdot 10^4$ and a test dataset of $10^4$ samples. All samples in MNIST datasets are $28 \times 28$ size gray-level images, and in CIFAR10 $32 \times 32$ color images (with three color channels each). We compare three alternatives: DP-SGD applied to a 2-layer ReLU network, DP-SGD applied to the stochastic convex model (3.6) without regularization ($\lambda = 0$), and NoisyCGD applied to the stochastic convex model (3.6) with $\lambda > 0$. We use the cross-entropy loss for all the models considered. To simplify the comparisons, we set the batch size to 1000 for all methods and train them for 400 epochs. We compare the results on two noise levels, $\sigma = 5.0$ and $\sigma = 15.0$, which correspond to privacy parameter $\varepsilon = 1.33$ and $\varepsilon = 4.76$ at $\delta = 10^{-5}$. In all experiments, the model parameters are always initialized to zero.

Although 2-layer networks with tempered sigmoid activations (Papernot et al., 2021) would likely yield improved results, we focus on ReLU networks as baselines for consistency. This allows us to compare the methods in the context of ReLU-based architectures and will not affect our main finding, which is that we can find improved models for the hidden state privacy analysis. Unlike Abadi et al. (2016), we do not use pre-trained convolutive layers for obtaining higher test accuracies in the CIFAR10 experiment, as we experimentally observe that the DP-SGD trained logistic regression gives similar accuracies as the DP-SGD trained ReLU network. We consider the much more difficult problem of training the models from scratch using the vectorized CIFAR10 images as features.

We also compare the hyperparameter tuning under differential privacy and describe this in Appendix G. The experimental results in this section use the hyperparameter from the private tuning process. The hyperparameter tuning of NoisyCGD is simplified by the fact that the bound of Thm. (2.7) depends monotonously on the parameter $c = 1 - \eta \cdot \lambda$. If the hyperparameters $b$ and noise scale $\sigma$ are fixed, fixing the GDP parameter $\mu$ also fixes the value of $c$. Thus, a grid of learning rate candidates also determines the values of $\lambda$'s. The hyperparameter grids are noted in Appendix G.1.

**Baselines.** As the stochastic approximation yields a convex problem, we compare, as a baseline, to a linear regression model that uses the same features. In contrast to the iterative methods, we use the Sufficient Statistics Perturbation method, which is a one-step solution (SSP Amin et al., 2023). SSP is a state of the art model for linear regression with Differential Privacy guarantees, and the details are provided in Appendix H. We consider three sets of features: a) the stochastic approximation features in the GLM formulation of Equation 3.7 (Convex approx.), b) the activation patterns of Equation 3.3 (Random ReLU), and c) Random Fourier Features of the input images Rahimi and Recht (2007).

Table 1: A comparison of model accuracies vs. $\varepsilon$-values. The iterative methods generally score better than SSP and have comparable accuracy, which shows a high-utility result for hidden-state privacy analysis of a 2-layer neural network. The results are the mean accuracy among five random restarts.

| | MNIST | | CIFAR-10 | |
|---|---|---|---|---|
| | $\varepsilon = 1.33$ | $\varepsilon = 4.76$ | $\varepsilon = 1.33$ | $\varepsilon = 4.76$ |
| Sufficient Statistics Perturbation (Convex Approx.) | $51.9_{\pm 1.1}$ | $67.0_{\pm 0.2}$ | $19.2_{\pm 1.1}$ | $23.3_{\pm 0.9}$ |
| Sufficient Statistics Perturbation (Random ReLU) | $52.5_{\pm 0.9}$ | $65.6_{\pm 0.3}$ | $19.3_{\pm 0.2}$ | $25.9_{\pm 0.3}$ |
| Sufficient Statistics Perturbation (RFF) | $64.1_{\pm 0.9}$ | $77.4_{\pm 0.2}$ | $21.3_{\pm 0.5}$ | $28.5_{\pm 0.3}$ |
| DP-SGD + Convex Approximation | $93.1_{\pm 0.1}$ | $94.9_{\pm 0.1}$ | $41.5_{\pm 0.2}$ | $45.5_{\pm 0.2}$ |
| DP-SGD + ReLU | $91.7_{\pm 0.1}$ | $94.3_{\pm 0.1}$ | $42.5_{\pm 0.1}$ | $47.0_{\pm 0.2}$ |
| NoisyCGD + Convex Approximation | $92.4_{\pm 0.2}$ | $94.4_{\pm 0.1}$ | $41.0_{\pm 0.2}$ | $45.4_{\pm 0.3}$ |

**Main Results.** Figures 2 and 3 show the accuracies of the models along a training of 400 epochs for the MNIST and CIFAR10 dataset. The results for FashionMNIST are in Appendix I. The figures also include the accuracies for the learning rate-optimized logistic regression models. We observe that the proposed convex model significantly outperforms logistic regression, which has been the most accurate model considered in the literature for hidden state DP analysis. The hyperparameters for this experiment were found with Differentially Private hyperparameter tuning (Appendix G).

Figure 2 shows that the convexification helps in the MNIST experiment: both DP-SGD and the noisy cyclic mini-batch GD applied to the stochastic dual problem lead to better utility models than DP-SGD applied to the ReLU network. Notice also that the final accuracies for DP-SGD are not far from the accuracies obtained by Abadi et al. (2016) using a three-layer network for corresponding $\varepsilon$-values which can be compared using the fact that there is approximately a multiplicative difference of 2 between the two relations: the add/remove neighborhood relation used by (Abadi et al., 2016) and the substitute neighborhood relation used in our work.

The results of Figures 2, 3 and Section I are averaged over five trials. The error bars on both sides of the mean values indicate 1.96 times the standard error. In all experiments, we set $\delta = 10^{-5}$.

The proposed method is also compared to non-iterative baseline methods. Table 1 compares NoisyCGD and DP-SGD against Sufficient Statistics Perturbation on random and non-linear features. For each dataset and privacy setting, even the best linear regression model has lower accuracy than any of the iterative methods. This shows that iterative methods for non-linear models can achieve higher accuracy, and we believe that NoisyCGD is an important step in this direction.

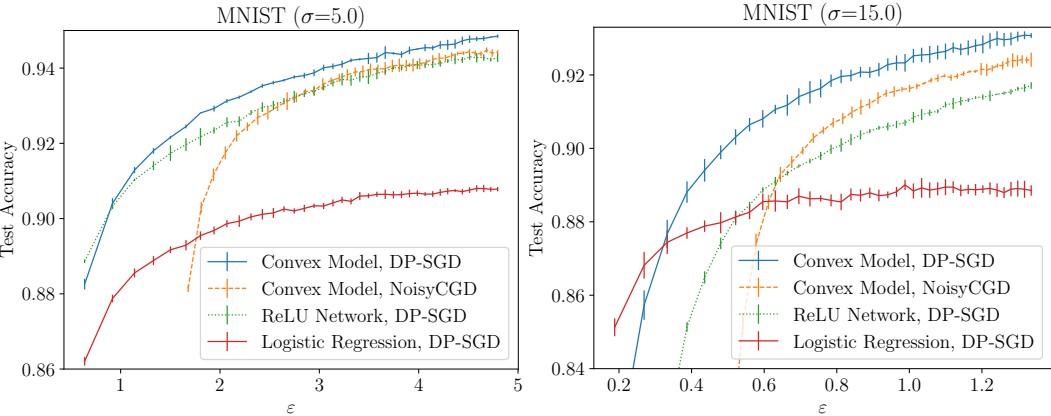

Figure 2: MNIST: Test accuracy versus the spent privacy budget $\varepsilon$, when each model is trained for 400 epochs. NoisyCGD and DP-SGD generally have comparable performance for the 2-layer ReLU network and much higher accuracy than logistic regression.

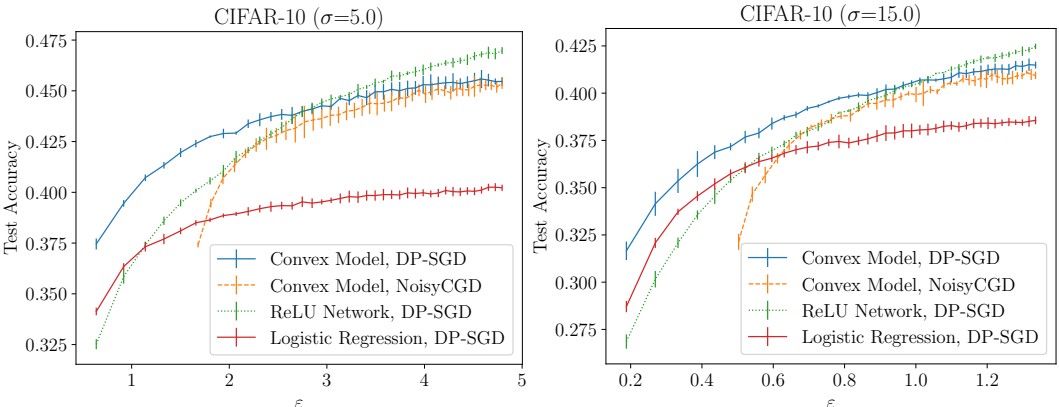

Figure 3: CIFAR10 Test accuracy versus the spent privacy budget $\varepsilon$, when each model is trained for 400 epochs. NoisyCGD and DP-SGD generally have comparable performance for the 2-layer ReLU network and much higher accuracy than logistic regression.

## 6 Conclusions and Outlook

We have shown how to privately approximate the two-layer ReLU network, and we have given the first high privacy-utility trade-off results using the hidden state privacy analysis. In particular, we have provided results for the noisy cyclic mini-batch GD, which is more suitable for practical applications of private ML model training than variants of DP-SGD that carry out random subsampling at each iteration. Experimentally, on benchmark image classification datasets, the results for the convex problems have similar privacy-utility trade-offs as those obtained by applying DP-SGD for a 2-layer ReLU network and using the composition analysis. An interesting task for future research is to obtain hidden state privacy guarantees for 'deeper' neural networks, for example, by developing convex models that approximate deeper neural networks, including those with convolutional layers (Ergen and Pilanci, 2020) and different activation functions (Ergen et al., 2023).

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

# A  Appendix

In the appendix, we provide detailed proofs and additional results that complement the main text. Section B contains the impact statement, and Section C clarifies the connection between the Strongly Convex Approximation and Generalized Linear Models. In Section D, we prove that the gradients of our loss function are Lipschitz continuous. In Section E, we provide a detailed convergence analysis to minimize the loss function within the random data model, and Section E.1 presents the reference algorithm for the utility bounds. Appendix F discusses the approximability of the stochastic approximation for the dual problem in the non-private case. Finally, Appendices G and H contain details on the hyperparameter tuning and experimental setting, respectively.

# B  Broader impact statement

Our work advances the differentially private machine learning field by providing improved privacy-utility trade-offs and theoretical understanding. This has positive societal implications as it enables training machine learning models while better protecting individual privacy. Differential privacy is increasingly important as machine learning systems process more sensitive personal data in healthcare, finance, and other domains.

However, it is important to acknowledge potential limitations and risks. Prior research has shown that differentially private training methods can disproportionately impact model performance on minority classes and underrepresented groups in the data (Farrand et al., 2020; Bagdasaryan et al., 2019). One hypothesis is that the additive noise tends to have a larger relative effect on patterns that appear less frequently in the training set. While our theoretical advances do not directly address this issue, we believe future work in DP must investigate techniques for ensuring fairness and equal utility across different subgroups.

Additionally, we note that improved privacy-utility trade-offs, while generally beneficial, could potentially promote more widespread adoption of machine learning in sensitive domains. The broader deployment should be approached carefully considering the societal implications and ethical guidelines for each application setting.

# C  Formulating the Strongly Convex Approximation as a GLM

We first show that the strongly convex loss function given in Eq. (3.6) corresponds to a loss function of a convex generalized linear model. The loss function in Eq. (3.6) is of the form

$$\mathcal{L}\big(v, X, y\big) = \frac{1}{n} \sum\nolimits_{j=1}^{n} \ell_j(v, x_j, y_j),$$

where

$$\ell_j(v, x_j, y_j) = \frac{1}{2} \left\| \sum_{i=1}^{P} (\Lambda_i)_{jj} x_j^\top v_i - y_j \right\|_2^2 + \frac{\lambda}{2} \sum_{i=1}^{P} \|v_i\|_2^2, \quad (\Lambda_i)_{jj} = \mathbb{1}(x_j^\top u_i \geq 0).$$

The $u_i$'s are the randomly sampled vectors that determine the boolean matrices $\Lambda_i$'s (and the functions $\ell_j$), and we denote the vector

$$v = \begin{bmatrix} v_1 \\ \vdots \\ v_P \end{bmatrix} \in \mathbb{R}^{P \cdot d}.$$

This is actually a generalized linear model: if we denote

$$\tilde{x}_j = \begin{bmatrix} (\Lambda_1)_{jj} x_j \\ \vdots \\ (\Lambda_P)_{jj} x_j \end{bmatrix},$$

we see that

$$\ell_j(v, x_j, y_j) = \frac{1}{2} \left\| \tilde{x}_j^\top v - y_j \right\|_2^2 + \frac{\lambda}{2} \|v\|_2^2,$$

which shows that we are actually minimizing a loss function of a GLM when we are minimizing the loss $\mathcal{L}(v, X, y)$ w.r.t. $v$. By the results of (Song et al., 2021), we know that the clipped gradients are gradients of an auxiliary convex loss, which allows using the privacy amplification by iteration analysis by Bok et al. (2024).

Moreover, the convexity properties of the GLM loss function are preserved under gradient clipping. This is shown in Appendix E.2 of (Redberg et al., 2024). In summary, we can use the convexity properties shown in our Section 3.4 for the privacy analysis.

## D   Proof of Lemma 3.1

**Lemma D.1.** *The gradients of the loss function*

$$\ell(v, x_j, y_j) = \frac{1}{2} \left\| \sum\nolimits_{i=1}^{P} (\Lambda_i)_{jj} x_j^\top v_i - y_j \right\|_2^2 + \frac{\lambda}{2} \sum\nolimits_{i=1}^{P} \|v_i\|_2^2$$

*are $\beta$-Lipschitz continuous for $\beta = \|x_j\|_2^2 + \lambda$.*

*Proof.* For the quadratic function

$$h(v) = \frac{1}{2} \left\| \sum\nolimits_{i=1}^{P} (\Lambda_i)_{jj} x_j^\top v_i - y_j \right\|_2^2$$

the Hessian matrix is of the diagonal block form

$$\nabla^2 h = \operatorname{diag}(A_1, \ldots, A_P),$$

where $A_i = x_j(\Lambda_i)_{jj} x_j^\top = x_j(\Lambda_i)_{jj} x_j^\top$. Since for all $i \in [P]$, $x_j(\Lambda_i)_{jj} x_j^\top \preccurlyeq x_j x_j^\top$, $\nabla^2 h \preccurlyeq x_j x_j^\top$ and furthermore for the spectral norm of $\nabla^2 h$ we have that $\left\| \nabla^2 h \right\|_2 \leq \left\| x_j x_j^\top \right\|_2 = \|x_j\|_2^2$. $\qquad \square$

## E   Utility Bound within the Random Data Model

### E.1   Reference Algorithm for the Utility Bounds

We consider the following form of DP gradient descent (DP-GD) for the theoretical utility analysis.

---

**Algorithm 1** Differentially Private Gradient Descent (Repeated from Song et al., 2021)

---

1: Input: dataset $D = \{z_i\}_{i=1}^n$, loss function $\ell : \mathbb{R}^p \times \mathcal{X} \to \mathbb{R}$, gradient $\ell_2$-norm bound $L$, constraint set $\mathcal{C} \subseteq \mathbb{R}^p$, number of iterations $T$, noise variance $\sigma^2$, learning rate $\eta$.
2: $\theta_0 \leftarrow 0$.
3: **for** $t = 0, \ldots, T-1$ **do**
4: $\quad g_t^{\text{priv}} \leftarrow \frac{1}{n} \sum\limits_{i=1}^{n} \partial_\theta \ell(\theta_t, z_i) + b_t$, where $b_t \sim \mathcal{N}(0, \sigma^2 I_d)$.
5: $\quad \theta_{t+1} \leftarrow \Pi_{\mathcal{C}} \left( \theta_t - \eta \cdot g_t^{\text{priv}} \right)$, where $\Pi_{\mathcal{C}}(v) = \operatorname{argmin}_{\theta \in \mathcal{C}} \|\theta - v\|_2$.
6: **end for**
7: **return** $\theta^{\text{priv}} = \frac{1}{T} \sum\limits_{t=1}^{T} \theta_t$.

---

We have the following classical result for Alg. 1.

**Theorem E.1** (Bassily et al. 2014; Talwar et al. 2014)**.** *If the constraining set $\mathcal{C}$ is convex, the data sample-wise loss function $\ell(\theta, z)$ is a convex function of the parameters $\theta \in \mathbb{R}^p$, $\|\nabla_\theta \ell(\theta, z)\|_2 \leq L$ for all $\theta \in \mathcal{C}$ and $z \in D = (z_1, \ldots, z_n)$, then for the objective function $\mathcal{L}(\theta, D) = \frac{1}{n} \sum_{i=1}^n \ell(\theta, z_i)$ under appropriate choices of the learning rate and the number of iterations in the gradient descent algorithm (Alg. 1), we have with probability at least $1 - \beta$,*

$$\mathcal{L}(\theta^{priv}, D) - \mathcal{L}(\theta^*, D) \leq \frac{L \|\theta_0 - \theta^*\|_2 \sqrt{p \log(1/\delta) \log(1/\beta)}}{n\varepsilon}.$$

## E.2 Utility Analysis

We give a convergence analysis for the minimization of the loss function (3.5) with $\lambda = 0$, i.e., our minimization problem is

$$\min_v h(v) = \frac{1}{2} \left\| \sum_{i=1}^{P} \Lambda_i X v_i - y \right\|_2^2, \tag{E.1}$$

where the diagonal boolean matrices $\Lambda_1, \ldots, \Lambda_P \in \mathbb{R}^{n \times n}$ are constructed by taking first $P$ i.i.d. samples $u_1, \ldots, u_P$, $u_i \sim \mathcal{N}(0, I_d)$, and then setting the diagonals $\Lambda_i$'s: $(\Lambda_i)_{jj} = \max\left(0, \operatorname{sign}(x_j^\top u_i)\right)$.

Kim and Pilanci (2024) have recently given several results for the minimization problem (3.4). The analysis uses the condition number $\kappa$ defined as

$$\kappa = \frac{\lambda_{\max}(XX^\top)}{\lambda_{\min}(\Sigma)},$$

where

$$\Sigma = \mathbb{E}_{g \sim \mathcal{N}(0, I_d)}[\operatorname{diag}[\mathbb{1}(Xg \geq 0)]XX^\top \operatorname{diag}[\mathbb{1}(Xg \geq 0)]]$$

and $\lambda_{\max}$ and $\lambda_{\min}$ denote the largest and smallest eigenvalues of a matrix, respectively. Kim and Pilanci (2024) provide the following lower bound on the number of random hyperplanes required, in terms of the condition number $\kappa$, for the problem (E.1) to attain a zero minimum.

**Lemma E.2** (Kim and Pilanci 2024, Proposition 2). *Suppose we sample $P \geq 2\kappa \log \frac{n}{\delta}$ hyperplane arrangement patterns and assume $M$ is invertible. Then, with probability at least $1 - \delta$, for any $y \in \mathbb{R}^n$, there exist $v_1, \ldots, v_P \in \mathbb{R}^d$ such that*

$$\sum_{i=1}^{P} \Lambda_i X v_i = y. \tag{E.2}$$

Furthermore, if we assume random data, i.e., $X_{ij} \sim \mathcal{N}(0, 1)$ i.i.d., then for sufficiently large $d$ we have the following bound for $\kappa$.

**Lemma E.3** (Kim and Pilanci 2024, Corollary 3). *Let the ratio $c = \frac{n}{d} \geq 1$ be fixed. For any $\gamma > 0$, there exists $d_1$ such that for all $d \geq d_1$ with probability at least $1 - \gamma - \frac{1}{(2n)^8}$,*

$$\kappa \leq 10\sqrt{2}\left(\sqrt{c} + 1\right)^2.$$

Combining Lemmas E.2 and E.2 gives us a high-probability lower bound for the number of hyperplanes in random data model, for the problem (E.1) to attain a zero minimum. Assuming the gradients stay bounded by a constant $L$, by applying the DP-ERM result of Thm. E.1 gives us the main result (Theorem 4.1 in the main paper):

**Theorem E.4.** *Consider applying the private gradient descent (Alg. 1) to the strongly convex loss (3.6) and assume that the gradients are bounded by a constant $L > 0$. Let the ratio $c = \frac{n}{d} \geq 1$ be fixed. For any $\gamma > 0$, there exists $d_1$ such that for all $d \geq d_1$, with probability at least $1 - \gamma - \frac{1}{(2n)^8}$ (with $\widetilde{O}$ omitting the logarithmic factors)*

$$\mathcal{L}(\theta^{priv}, D) \leq \widetilde{O}\left(\frac{1}{\sqrt{n}\varepsilon}\right).$$

*Proof.* From Lemmas E.2 and E.3 it follows that by taking $d$ and $n$ large enough (such that $n \geq d$), we have that with $P = O(\frac{n \log \frac{n}{\gamma}}{d})$ hyperplane arrangements there exists $u \in \mathbb{R}^{d \cdot P}$ such that Eq. (E.2) holds with probability at least $1 - \gamma - \frac{1}{(2n)^8}$. Applying Thm. E.1 with the ambient dimension $p = d \cdot P = O(n \log \frac{n}{\gamma})$ then shows the claim. $\qquad \square$

# F   Illustrations of the Stochastic Approximation

## F.1   Illustration with SGD Applied to MNIST

Figure 4 illustrates the approximability of the stochastic approximation for the dual problem in the non-private case, when the number of random hyperplanes $P$ is varied, for the MNIST classification problem described in Section 5. We apply SGD with batch size 1000 to both the stochastic dual problem and a fully connected ReLU network with hidden-layer width 200. For each model, we optimize the learning rate using the grid $\{10^{-i/2}\}$, $i \in \mathbb{Z}$. This comparison shows that the approximability of the stochastic dual problem increases with increasing $P$.

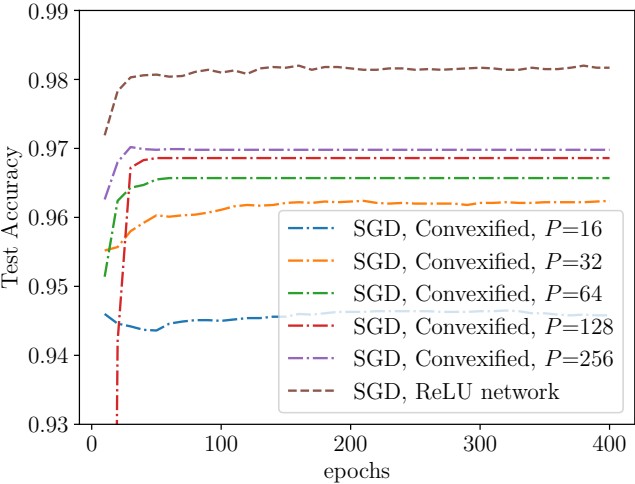

Figure 4: Test accuracies vs. number of epochs, when all models are trained using SGD with batch size 1000. The number of random hyperplanes $P$ varies for the stochastic dual problem. The ReLU network is a 2-layer fully connected ReLU network with a hidden-layer width of 200. Cross-entropy loss is used for all models.

## F.2   Illustration with DP-SGD Applied to MNIST

Figure 5 illustrates the approximability of the stochastic approximation for the dual problem in the private case, when the number of random hyperplanes $P$ is varied, for the MNIST classification problem described in Section 5. We apply DP-SGD with batch size 1000 to both the stochastic dual problem and to a fully connected ReLU network with hidden-layer width 200, and for each model, optimize the learning rate using the grid $\{10^{-i/2}\}$, $i \in \mathbb{Z}$. Based on these comparisons, we conclude that $P = 128$ is not far from optimum, as increasing the dimension means that the adverse effect of the DP noise becomes larger.

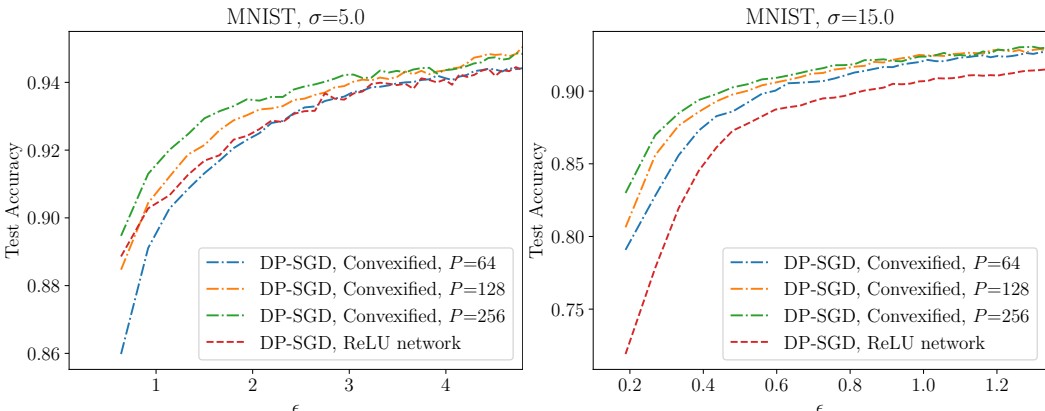

Figure 5: Test accuracies vs. number of epochs, when all models are trained using DP-SGD with batch size 1000, for two different noise levels $\sigma$. The number of random hyperplanes $P$ varies for the stochastic dual problem. The ReLU network is a 2-layer fully connected ReLU network with hidden-layer width 200.

# G   DP Hyperparameter Tuning for DP-SGD and NoisyCGD

When comparing DP optimization methods empirically, it is also important to consider the effect of the hyperparameter tuning on the privacy costs. Most relevant to our work are the private selection methods given by Liu and Talwar (2019) and Papernot and Steinke (2022) that are applicable to private hyperparameter tuning and the privacy profile-based analysis of those methods by Koskela et al. (2024). These results apply to tuning algorithms that return the best model of $\mathcal{K}$ randomly chosen alternatives, where $\mathcal{K}$ is also a random variable. We consider the case where $\mathcal{K}$ is Poisson distributed. However, other alternatives exist that allow adjusting the balance between compute cost of training, privacy and accuracy (Papernot and Steinke, 2022; Koskela et al., 2024). The DP bounds for Poisson distributed $\mathcal{K}$ can be described as follows. Let $Q(y)$ be the density function of the quality score of the base mechanism ($y$ denoting the score) and $A(y)$ be the density function of the tuning algorithm that outputs the best model of the $\mathcal{K}$ alternatives. Let $A$ and $A'$ denote the output distributions of the tuning algorithm evaluated on neighboring datasets $D$ and $D'$, respectively. Then, the hockey-stick divergence between $A$ and $A'$ can be bounded using the following result.

**Theorem G.1** (Koskela et al. 2024). *Let $\mathcal{K} \sim \text{Poisson}(m)$ for some $m \in \mathbb{N}$, and let $\delta(\varepsilon_1)$, $\varepsilon_1 \in \mathbb{R}$, define the privacy profile of the base mechanism $Q$. Then, for all $\varepsilon > 0$ and for all $\varepsilon_1 \geq 0$,*

$$H_{\mathrm{e}^\varepsilon}(A||A') \leq m \cdot \delta(\widehat{\varepsilon}), \tag{G.1}$$

*where $\widehat{\varepsilon} = \varepsilon - m \cdot (\mathrm{e}^{\varepsilon_1} - 1) - m \cdot \delta(\varepsilon_1)$.*

Theorem G.1 says that the hyperparameter tuning algorithm is $\left(\widehat{\varepsilon}, m \cdot \delta(\widehat{\varepsilon})\right)$-DP in case the base mechanism is $(\varepsilon_1, \delta(\varepsilon_1))$-DP. If we can evaluate the privacy profile for different values of $\varepsilon$, we can also optimize the upper bound (G.1). When comparing DP-SGD and NoisyCGD, we use the fact that DP-SGD privacy profiles are approximately those of the Gaussian mechanism for large compositions (Sommer et al., 2019) as follows. Suppose DP-SGD is $(\varepsilon^*, \delta^*)$-DP for some values of the batch size $b$ noise scale $\sigma$ and number of iterations $T$. Then, we fix the value of the constant $c = \eta \cdot \lambda$ for NoisyCGD such that it is also $(\varepsilon^*, \delta^*)$-DP for the same hyperparameter values $b$, $\sigma$ and $T$, giving some GDP parameter $\mu^*$. Along the GDP privacy profile determined by $\mu^*$, we find $(\varepsilon_1, \delta(\varepsilon_1))$ that optimizes the bound of Eq. (G.1), and then evaluate the DP-SGD-$\delta$ using that same value $\varepsilon_1$, giving some value $\widehat{\delta}(\varepsilon_1)$. Taking the maximum of $\delta(\varepsilon_1)$ and $\widehat{\delta}(\varepsilon_1)$ for the evaluation of $\widehat{\varepsilon}$ in the bound of Eq. (G.1) will then give a privacy profile that bounds the DP-guarantees of the hyperparameter tuning of both DP-SGD and NoisyCGD.

Overall, in case the batch size, number of epochs and $\sigma$ are fixed, in addition to the learning rate $\eta$, we have in all alternatives only one hyperparameter to tune: the hidden-layer width $W$ for the ReLU networks and the number of hyperplanes $P$ for the convex models.

Table 2: Model accuracies vs. $\varepsilon$-values for the DP hyperparameter tuning algorithm. The number of candidate models $\mathcal{K}$ is Poisson distributed with mean $m = 20$. The iterative methods generally have comparable accuracy and score better than SSP – similar to the conclusions from Table 1.

|  | MNIST | | CIFAR-10 | |
|---|---|---|---|---|
|  | $\varepsilon = 2.88$ | $\varepsilon = 8.91$ | $\varepsilon = 2.88$ | $\varepsilon = 8.91$ |
| Sufficient Statistics Perturbation (Convex Approx.) | $62.8_{\pm.4}$ | $71.9_{\pm.1}$ | $21.7_{\pm1.0}$ | $24.5_{\pm.7}$ |
| Sufficient Statistics Perturbation (Random ReLU) | $61.7_{\pm.4}$ | $68.3_{\pm.1}$ | $23.2_{\pm.5}$ | $28.5_{\pm.4}$ |
| Sufficient Statistics Perturbation (RFF) | $73.5_{\pm.5}$ | $80.1_{\pm.3}$ | $25.6_{\pm.3}$ | $31.1_{\pm.1}$ |
| DP-SGD + Convex Approximation | $92.8_{\pm.3}$ | $94.8_{\pm.2}$ | $41.6_{\pm.2}$ | $45.6_{\pm.3}$ |
| DP-SGD + ReLU | $91.6_{\pm.3}$ | $94.1_{\pm.2}$ | $42.5_{\pm.2}$ | $47.2_{\pm.2}$ |
| NoisyCGD + Convex Approximation | $92.4_{\pm.3}$ | $94.3_{\pm.2}$ | $41.1_{\pm.2}$ | $45.6_{\pm.3}$ |

### G.1 Hyperparameter Grids Used for the Experiments

The hyperparameter grids for the number of random hyperplanes $P$ for the convex model and the hidden width $W$ for the ReLU network are chosen based on the GPU memory of the available machines.

The learning rate $\eta$ is tuned in all alternatives using the grid

$$\{10^{-3.0}, 10^{-2.5}, 10^{-2.0}, 10^{-1.5}, 10^{-1.0}, 10^{-0.5}\}. \tag{G.2}$$

For MNIST and FashionMNIST, we tune the number of random hyperplanes $P$ over

$$\{32, 64\}$$

and for CIFAR10 over

$$\{16, 32, 64\}.$$

For MNIST and FashionMNIST, we tune the hidden width $W$ of the ReLU network over

$$\{200, 500, 800\}$$

and for CIFAR10 over

$$\{200, 400\}.$$

### G.2 Results on DP Hyperparameter Tuning

Table 2 shows the accuracies of the best models obtained using the DP hyperparameter tuning algorithm. With the noise scale values $\sigma = 5.0$ and $\sigma = 15.0$, the DP-SGD trained models are $(1.33, 10^{-5})$-DP and $(4.76, 10^{-5})$-DP, respectively. To have similar privacy guarantees for the base models trained using NoisyCGD, we adjust the regularization constant $\lambda$ accordingly (as depicted in Section G), which leads to equal final $(\varepsilon, \delta)$-DP guarantees for the hyperparameter tuning algorithms. We see from Tables 2 that the convex models are on par in accuracy with the ReLU network. Notice also from the results of Section 5 that the learning rate tuned logistic regression cannot reach similar accuracies as the NoisyCGD trained convex model.

The results of Table 2 were the most computationally intensive part of our experiments and were run on eight NVIDIA GeForce RTX 3090 GPUs for approximately 3 days each.

### G.3 Batch Size Ablation

To justify the choice of batch size 1000 for all experiments of Section 5, we consider an ablation experiment for MNIST where we study the effect of the batch size.

Figures 6 and 7 show the effect of the batch size on the final test accuracy for three alternatives, when we train all models for 400 epochs. Each point is an average of three runs, and the learning rate for each model is optimized using the grid (G.2).

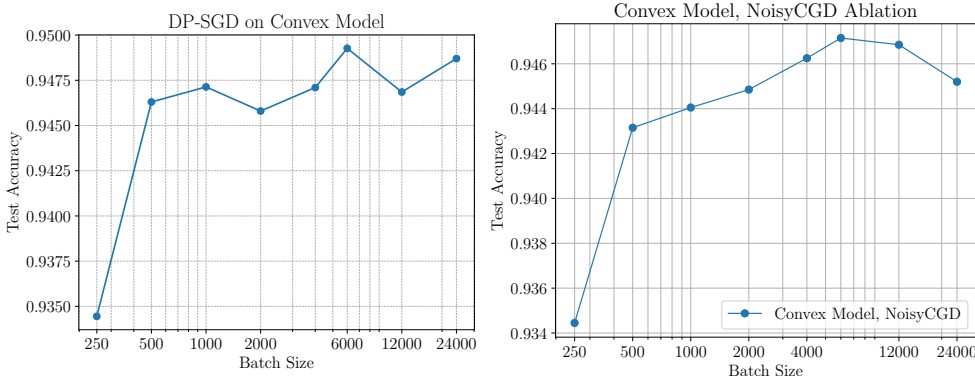

Figure 6: Ablation of batch size for MNIST: test accuracy vs. batch size with $\delta = 10^{-5}$ and 400 training epochs. Left: DP-SGD on the convexified model. Right: NoisyCGD on the convexified model.

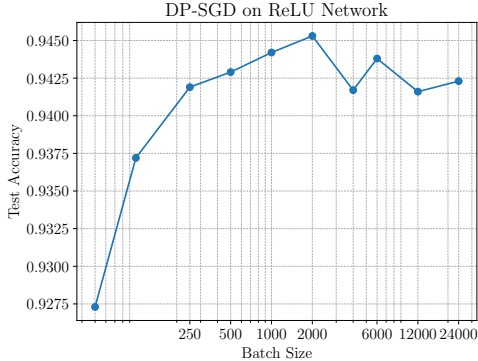

Figure 7: MNIST Comparisons: Test accuracies vs. batch size for DP-SGD applied on a ReLU network with hidden layer width 500, when $\delta = 10^{-5}$ and each model is trained for 400 epochs.

# H Experimental Details for Baselines

## H.1 Sufficient Statistics Perturbation (SSP)

We compare the proposed method with private linear regression using Sufficient Statistics Perturbation (SSP) from Amin et al. (2023). SSP is a one-step method that computes and perturbs the sufficient statistics without requiring iterations. This has the advantage that computation only scales linearly with the number of samples and cubically with the number of features, i.e., for calculating the inverse of the sample covariance matrix. The method clips input features adaptively and adds noise to the sufficient statistics as part of achieving differential privacy. Following (Wang, 2018), we estimate the clipping norm directly from data rather than setting it ahead of time. This simplification yields optimistic results compared to pre-determined clipping – in theory, one should set the clipping norm ahead of time. However, it allows direct evaluation of the baseline experiment across datasets without introducing another hyperparameter. This does not affect our conclusions, since no result from SSP outperforms the iterative methods in our experiments.

All the datasets are multi-class classification problems, and we run multiple one-versus-all linear regressions. For example, for MNIST with 10 classes, we run the SSP method ten times – one for each class, by predicting 1 for the specific class and 0 for all other classes. The number of features is the same as the number of features in the other comparing method. This means that for MNIST and FashionMNIST, there are either 100 random hyperplanes for the ReLU methods, or 100 random Fourier features. For CIFAR10, we use only 64 random features for comparability with the iterative methods which have computational constraints. The SSP has one hyperparameter (parameter $\rho$ in Amin et al., 2023) and we set it to the recommended default value of 0.05.

### H.2 Logistic Regression with DP-SGD

Previous work shows that DP-SGD applied to logistic regression combined with composition analysis leads to similar privacy-utility trade-offs as Noisy cyclic GD comgined with the hidden-state analysis (Bok et al., 2024). To this end, we compare against DP-SGD applied to logistic regression in our experiments, and provide the details of the experiment in this section.

The logistic regression model uses softmax classification for multi-class prediction, defined as

$$f(x)_j = \frac{e^{w_j^\top x + b_j}}{\sum_{k=1}^{K} e^{w_k^\top x + b_k}}, \tag{H.1}$$

where $f(x)_j$ is the predicted probability for class $j$, $w_j \in \mathbb{R}^d$ is the weight vector for class $j$, $b_j \in \mathbb{R}$ is the bias term for class $j$, and $K$ is the number of classes. The model parameters are trained jointly using the cross-entropy loss

$$\ell(W, b, x, y) = -\sum_{j=1}^{K} y_j \log(f(x)_j), \tag{H.2}$$

where $y \in \{0, 1\}^K$ is the one-hot encoded label vector, $W = [w_1, \ldots, w_K]^\top \in \mathbb{R}^{K \times d}$ is the weight matrix containing all class weight vectors, and $b = [b_1, \ldots, b_K]^\top \in \mathbb{R}^K$ contains all bias terms.

## I   Additional Experimental Results on FashionMNIST

Figure 8 shows the accuracies of the best models along the training iteration of 400 epochs for the FashionMNIST experiment.

For comparison, we train a linear regression model with SSP. At $\varepsilon = 4.76$, this results in $67.9_{\pm0.4}$ and $73.2_{\pm0.3}\%$ accuracy for Random ReLU and RFF, respectively. At $\varepsilon = 1.33$, the baseline accuracies are $59.9_{\pm1.0}$ and $66.4_{\pm1.1}\%$, respectively.

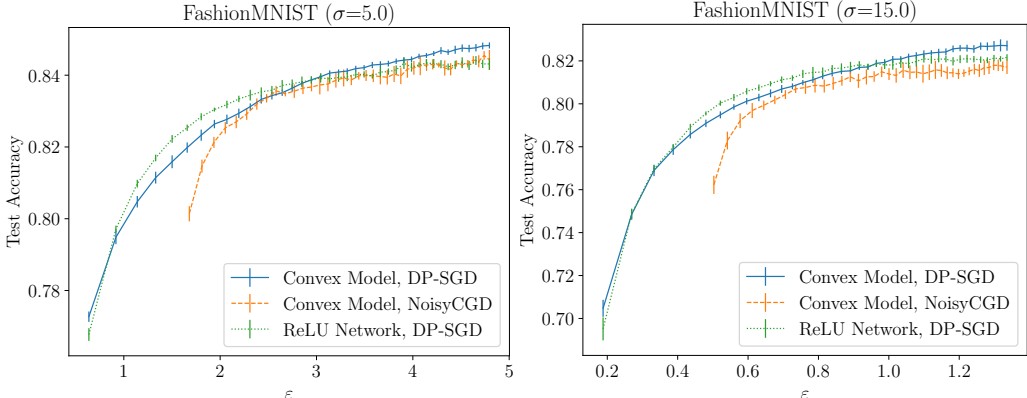

Figure 8: FashionMNIST: Test accuracies vs. the spent privacy budget $\varepsilon$, when each model is trained for 400 epochs. The model is a 2-layer ReLU network. Similar to Figures 2 and 3, NoisyCGD and DP-SGD achieve comparable accuracies and score a lot higher than baseline regression models.

## J Further Motivation for NoisyCGD Analysis: Evaluation of Shuffling Amplification Bounds

When using disjoint batches of data, currently the best option for obtaining rigorous guarantees is to use shuffling amplification (Feldman et al., 2021). However, as shown by Chua et al. (2024a,b), the data shuffling combined with disjoint batches leads to an inferior privacy-utility trade-off compared to random mini-batch sampling. And, as we experimentally show that the method we propose (strongly convex approximation of ReLU problem + NoisyCGD) has a similar privacy-utility trade-off as random mini-batch sampling applied to 2-layer ReLU networks, we believe that our approach would be superior compared to the shuffling approach.

To illustrate the $(\varepsilon, \delta)$-DP shuffling amplification bounds of (Thm. 3.8 Feldman et al., 2021) for DP-SGD with disjoint mini-bathces, we consider a setting in one of our experiments where we use noise parameter $\sigma = 5.0$. Similarly to the experiments of Chua et al. (2024b), we use the numerical method presented in Feldman et al. (2021) to accurately compute the shuffling bounds. Further, we combine the bounds off (Thm. 3.8 Feldman et al., 2021) with the improved bounds of (Thm. 3.1 Feldman et al., 2023)

In our experiments of Section 5, we use batch size of 1000 translating to 50 or 60 disjoint batches per epoch. For computing the shuffling bounds using (Thm. 3.8 Feldman et al., 2021), one can see that this is too small number of batches for the conditions of the analysis to hold. We also see that the shuffling privacy guarantee clearly improves as the number of batches per epoch grows (see, e.g., the comparisons by Chua et al., 2024b). To obtain a lower bound for the $(\varepsilon, \delta)$-DP upper bound, we consider 1000 batches per epoch instead.

The comparison to the bounds of the Gaussian mechanism (i.e., without any amplification, using parallel composition) is depicted in Fig. 9 which shows that the privacy guarantees of shuffling are worse than the privacy bounds of the Gaussian mechanism, which indicates that the privacy-utility trade-offs would be inferior when using data shuffling and shuffling amplification for the DP guarantees when using state-of-art amplification bounds for shuffling.

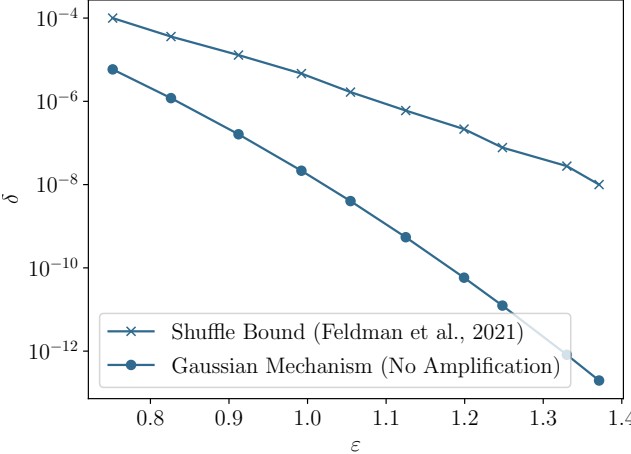

Figure 9: $(\varepsilon, \delta)$-DP guarantees for a single epoch of training when using 1000 disjoint batches and noise parameter $\sigma = 5.0$ obtained using the shuffling amplification of (Thm 3.8, Feldman et al., 2021). In experiments, we use 50 or 60 batches per epoch, in which case the DP guarantees of the shuffling would be even worse.

# K  Summary of Notation

- $d$: feature dimensionality, i.e., each data feature $x \in \mathbb{R}^d$;
- $n$: number of training samples (introduced in Section 2);
- $X \in \mathbb{R}^{n \times d}$: matrix of feature vectors, i.e., $X^\top = [x_1 \quad \ldots \quad x_n]$;
- $y \in \mathbb{R}^n$: vector of labels;
- $\ell(v, x_j, y_j)$: sample-wise loss;
- $\mathcal{L}(v, X, y)$: loss function for parameter vector $v$, data matrix $X$ containing examples $x_j$, and label vector $y$ with entries $y_j$;
- $\delta$: parameter in the definition of DP, set to $10^{-5}$ in all experiments;
- $\varepsilon$: parameter in the definition of DP;
- $D \sim D'$: two adjacent datasets (datasets are adjacent if they differ in one sample);
- $\mu$: privacy parameter in Gaussian DP;
- $E$: number of training epochs;
- $B_j$: mini-batch at iteration $j$;
- $\beta$: smoothness parameter (Lemma 3.1); a function $f$ is $\beta$-smooth if $\nabla f$ is $\beta$-Lipschitz;
- $C > 0$: clipping constant; $\mathrm{clip}(\cdot, C)$ denotes the clipping function that limits gradient 2-norms to at most $C$;
- $\eta$: learning rate;
- $H_\alpha(P\|Q)$: hockey-stick divergence between distributions $P$ and $Q$;
- $K$: number of classes in the multi-class model;
- $\kappa$: condition number used in the convex approximation analysis;
- $f : \mathbb{R}^d \to \mathbb{R}$: one-layer MLP mapping $d$-dimensional input features to a scalar prediction;
- $\lambda$: parameter of strong convexity; a function $f(\cdot)$ is $\lambda$-strongly convex if $g(x) = f(x) - \frac{\lambda}{2}\|x\|_2^2$ is convex; with overloaded notation, $\lambda_{\max}$ and $\lambda_{\min}$ denote the largest and smallest eigenvalues of a matrix;
- $\Lambda_i$: diagonal Boolean matrix representing the $i$th ReLU activation pattern;
- $L$: gradient sensitivity (Definition 2.6);
- $\|\cdot\|$: Euclidean norm;
- $\mathbb{1}(\zeta \geq 0)$: Iverson bracket for $(\zeta \geq 0)$: equal to 1 if $\zeta \geq 0$ and 0 otherwise; applied elementwise when $\zeta$ is a vector;
- $M = |\mathcal{D}_X|$: number of possible ReLU activation patterns in a two-layer ReLU network, i.e., the number of regions in a partition of $\mathbb{R}^d$ by hyperplanes through the origin perpendicular to the rows of $X \in \mathbb{R}^{n \times d}$;
- $\widetilde{O}(\cdot)$: Notation for Big-O that omits logarithmic factors;
- $P$: number of randomly sampled normal vectors (hyperplanes) in the convex approximation of the ReLU network, or number of Fourier components in the RFF model;
- $\Pi_{\mathcal{C}}(v) = \arg\min_{\theta \in \mathcal{C}} \|\theta - v\|_2$: projection of vector $v$ onto the set $\mathcal{C}$;
- $r = \mathrm{rank}(X)$: rank of the matrix $X$;
- $v$: vector of trainable parameters in the convex approximation;
- $v_i, w_i \in \mathbb{R}^d$: learnable $d$-dimensional vector lying in cone $\mathcal{V}_i$;
- $Z_j \sim \mathcal{N}(0, \sigma^2 I_d)$: Gaussian noise vector.

