# OpenReview forum: "Convex Approximation of Two-Layer ReLU Networks for Hidden State Differential Privacy"
_NeurIPS.cc/2025/Conference — NeurIPS 2025 poster_

### Official Review · Reviewer_5hye · 2025-06-24

**Clarity:** 4
**Significance:** 3
**Originality:** 3
**Rating:** 5
**Confidence:** 4

**Summary:**

Summary
This paper studies a scenario in which the adversary only has access to the final model and asks whether two-layer ReLU networks can be trained with strong hidden-state DP guarantees. The approach proceeds in three main steps:

1. Convex reformulation

Using Pilanci \& Ergen's dual perspective, the two-layer ReLU objective is rewritten as a convex program that enumerates all activation patterns. For sufficiently wide hidden layers, this convex formulation attains the same optimum as the original nonconvex problem.

2. Stochastic, strongly convex approximation

The intractable full pattern set is replaced by $P \ll M$ random hyperplanes, and an $\ell_2$ regularization term is added. This yields the smooth, $\lambda$-strongly convex objective in (3.6)-(3.7).

3. Noisy Cyclic Gradient Descent (NoisyCGD)

The convex approximation is trained using fixed, disjoint mini-batches with added Gaussian noise. Privacy loss is tracked via Bok et al.'s $\mu$-GDP accountant.

**Questions:**

Questions

1. Line 194: Equation (3.2) includes a regularization term. Is this added specifically to prepare for the strongly convex approximation that follows?

2. Line 202: Why does the bound on the matrices $D_X$ depend only on $\operatorname{rank}(X)$ and not on the parameter $u$, given the definition of $D_X$ ?

3. Line 205: Is equation (3.4) the dual form of (3.2)? It would be helpful to include a brief explanation of the roles of $v_i$ and $w_i$.

4. Line 277 (Section 4.1): What does the constant $c$ represent? Additionally, what are the implications in the high-dimensional regime (when $n \ll d$ )?

**Ethical Concerns:**

["NO or VERY MINOR ethics concerns only"]

**Final Justification:**

My concerns have been addressed, and thus I would like to increase my score to confirm acceptance.

**Quality:**

4

**Strengths And Weaknesses:**

Strengths

1. The authors tackle an important DP setting-where the attacker observes only the final model-and develop both theoretical and practical tools under convexity assumptions. Their techniques successfully extend to two-layer ReLU networks.

2. Mapping a two-layer ReLU network to a convex program and integrating it with DP yields an improved privacy-utility trade-off. This insight is both novel and significant.

3. The writing is well structured, and the related work section is comprehensive and detailed.

Weaknesses:

I did not identify any major flaws; overall, I find the work interesting and well executed. However, I have a few questions and suggestions for clarification:

---

> ### Author Rebuttal · Authors · 2025-07-30
>
> We thank the reviewer for identifying our paper as studying "an important setting" in privacy-preserving machine learning and the insights given by our paper "both novel and significant".
>
> Here are our answers to the questions:
>
> >Line 194: Equation (3.2) includes a regularization term. Is this added specifically to prepare for the strongly convex approximation that follows?
>
> Yes, the $L_2$-regularized ReLU problem of Eq. (3.2) is part of the motivation: for sufficiently wide ReLU networks, that problem is equivalent to the convex problem given Eq. (3.4) that has the $\ell_2$ - $\ell_1$ - regularization (so-called group lasso regularization). We make further approximations, including changing the $\ell_2$ - $\ell_1$ - regularization to $L_2$-regularization, which ensures strong convexity of the final approximation given in Eq. (3.6).
>
> > Line 202: Why does the bound on the matrices $D_X$ depend only on $\operatorname{rank}(X)$ and not on the parameter $u$, given the definition of $D_X$ ?
>
> Notice that $\mathcal{D}_X$ defined in Eq. (3.3) is the set of all possible boolean matrices, where each boolean matrix corresponds to a different pattern of ReLU activations for the features of the dataset (the rows of the matrix $X \in \mathbb{R}^{n \times d}$). Thus, the bound for $| \mathcal{D}_X |$ on line 202 is a bound for the number of possible hyperplane arrangements, and thus it is only determined by the feature matrix $X$. We will add 1-2 clarifying sentences to this part in the updated version of the paper.
>
> > Line 205: Is equation (3.4) the dual form of (3.2)? It would be helpful to include a brief explanation of the roles of $v_i$ and $w_i$.
>
> Thank you for the suggestion - we will also clarify this more in the updated paper. Since we remove the constraints for $v_i$ and $w_i$, we can merge them together into the variables $v_i$ in the final approximation given in Eq. (3.6).
>
> > Line 277 (Section 4.1): What does the constant $c$ represent? Additionally, what are the implications in the high-dimensional regime (when $n \ll d$)?
>
> We thank the reviewer for this valuable comment. Let us look at this in more detail.
>
> Constant $c$ equals $n/d$, where $n$ is the number of data points and $d$ is the feature dimension. We assume that the ratio $n/d$ stays constant in order to be able to use the results of [1]. One motivation for the assumption $c > 1$ comes from having a similar setting as in prior work: as we state in the last sentence before Thm. 4.1, we make an explicit comparison to the existing bounds in the random data model for the DP-SGD trained linear regression [1], where the dependency on the parameter dimension p appears. In the baseline work [1] it is assumed that $n = \Omega (d)$ (see e.g. Thm. 1.2 of [1]), which is implied by the assumption $n/d \geq 1$. The results are in this sense comparable in case $n/d \geq 1$ (with the assumption $n/d < 1$ there is no implication in any direction, we cannot compare them). As our Thm. 4.1 indicates, it is possible to get rid of the dependency parameter dimension on $p$ when training our proposed model with DP-SGD, and in this sense, our proposed model is better in the random data model than the private linear regression analyzed in [1].
>
> Another motivation, or rather requirement, comes from the mathematical analysis: the current analysis of [2] that is applicable to our proposed model has to assume $n/d \geq 1$ for the model loss to have, with high probability, zero global minimum for every possible label vector y with a limited number of random hyperplanes $P = O\big((n \log n) / d\big)$. This seems to be mainly a limitation of the analysis. If we assume $n/d < 1$, in a sense, the problem becomes easier since the feature matrix $X$ becomes column rank deficient with high probability. Then, there also might be an infinite number of global minimizers. Also, the DP-SGD convergence results [3] and [4], which we combine with the analysis of our proposed model, do not require a unique global minimizer. While the more difficult case $n/d \geq 1$ might be more interesting, it is an interesting question how to extend the results to $n/d < 1$. We will mention this as a remark after Thm. 4.1 in the updated version of the paper.
>
>
> [1] [Brown, G., Dvijotham, K., Evans, G., Liu, D., Smith, A., & Thakurta, A. Private Gradient Descent for Linear Regression: Tighter Error Bounds and Instance-Specific Uncertainty Estimation. ICML 2024.](https://arxiv.org/abs/2402.13531)
>
> [2] [Kim, S., & Pilanci, M. Convex Relaxations of ReLU Neural Networks Approximate Global Optima in Polynomial Time. ICML 2024.](https://arxiv.org/abs/2402.03625)
>
> [3] [Bassily, R., Smith, A., & Thakurta, A. Private empirical risk minimization: Efficient algorithms and tight error bounds. FOCS 2014.](https://arxiv.org/abs/1405.7085)
>
> [4] [Talwar, K., Thakurta, A., & Zhang, L. (2014). Private empirical risk minimization beyond the worst case: The effect of the constraint set geometry. arXiv.](https://arxiv.org/abs/1411.5417)

---

> > ### Comment · Reviewer_5hye · 2025-08-04
> > **Confirming acceptance**
> >
> > Thank you to the authors for the detailed responses. My concerns have been addressed, and I will increase my score and confirm acceptance.

---

### Official Review · Reviewer_k3aB · 2025-07-02

**Clarity:** 4
**Significance:** 3
**Originality:** 3
**Rating:** 4
**Confidence:** 4

**Summary:**

The paper studies the hidden-state DP threat model, which has been limited to the Logistic Regression model. In particular, the work studies the threat model for the 2-layer ReLU network by introducing a stochastic approximation of a dual formulation of the ReLU minimization problem. The analysis helps further analysis of the NoisyCGD defense method.

**Questions:**

See weaknesses.

**Ethical Concerns:**

["NO or VERY MINOR ethics concerns only"]

**Final Justification:**

After consideration, I've decided to keep my rejection decision.

**Limitations:**

Yes

**Quality:**

3

**Strengths And Weaknesses:**

Strengths:
1) The paper is well-written: well-defined problem, clear prelims with rigorous theoretical results.
2) While the experiments are limited to MNIST and CIFAR10 only, they are well designed. I also find the results reasonable and supportive toward the theoretical claims.
3) I find the studied problem of hidden-state privacy is interesting and of great interests.

Weaknesses:
I find the practical implication of the theoretical analysis is limited: (1) while the authors showcase the usage of the theoretical analysis in NoisyCGD, I find that implication is niche and is not strictly beneficial (I find NoisyCGD is actually significantly lower in performance compared to DP-SGD, not "on par" as the authors stated). (2) I understand that improving the analysis from logistic regression to 2-layer ReLU is novelty, but the results are still very far-away from practicality for real-world neural networks. How is the analysis look like for N-layer networks?

---

> ### Author Rebuttal · Authors · 2025-07-30
>
> We thank the reviewer for highlighting our "clear prelims," "rigorous theoretical results," and "well designed" experiments.
>
> DP learning of neural networks is important with the growing adoption of deep learning. With DP-SGD and variants, the options are limited, and there is a big need for a hidden-state analysis of the privacy loss. There are no results for $N$-layer networks yet, but we believe that this paper is an important first step in this direction. Also, we think that the steps taken to convexify a 2-layer network are non-trivial and deserve their own study. Two-layer ReLU networks can represent a much richer class of models than logistic regression, allowing them to solve a much broader variety of problems.
>
> On niche of NoisyCGD: there are several recent papers in top venues addressing the practical limitations of the random mini-batch DP-SGD, which is the predominant algorithm used for training ML models under DP constraints. For example, the line of research on so-called shufflers [1] aims to remove the need for random mini-batches, while recent works [2,3] compare its utility against random mini-batch data sampling in gradient-based DP model training. Several recent works tackle the privacy analysis of DP-SGD with random-sized disjoint mini-batches [4,5,6] obtained from randomly shuffled data. However, none of the above papers are able to give high-utility models trained with disjoint equal-sized mini-batches of data – something that would be very useful in practical implementations. This is also addressed in our Introduction: “...existing analyses do not accommodate fixed-size batches and in particular unshuffled data, further motivating our approach, which yields high-utility convex models for which NoisyCGD can be analyzed accurately.” We believe there is great practical value for the large-scale setting in being able to train the models with pre-defined disjoint batches, as is also expressed in the introduction of the paper [3]: “In practice, almost all deep learning systems generate mini-batches of fixed-size by sequentially going over the dataset, possibly applying a global shuffling of all the examples in the dataset for each training epoch; each epoch corresponds to a single pass over the dataset, and the ordering of the examples may be kept the same or resampled between different epochs. However, performing the privacy analysis for such a mechanism has appeared to be technically difficult due to the correlation between the different mini-batches.”
>
>
> [1] [Feldman, McMillan, and Talwar. Stronger privacy amplification by shuffling for Rényi and approximate differential privacy. In Proceedings of the Annual Symposium on Discrete Algorithms (SODA 2023).](https://arxiv.org/abs/2208.04591)
>
> [2] [Chua, Ghazi, Kamath, Kumar, Manurangsi, Sinha, and Zhang. How private are DP-SGD implementations? ICML 2024.](https://arxiv.org/abs/2403.17673)
>
> [3] [Chua, Ghazi, Kamath, Kumar, Manurangsi, Sinha, and Zhang. Scalable DP-SGD: Shuffling vs. poisson subsampling. NeurIPS 2024.](https://arxiv.org/abs/2411.04205)
>
> [4] [Feldman and Shenfeld  (2025). Privacy amplification by random allocation. arXiv.](https://arxiv.org/abs/2502.08202)
>
> [5] [Chua, Ghazi, Harrison, Kamath, Kumar, Leeman, Manurangsi, Sinha, and Zhang. Balls-and-bins sampling for DP-SGD. AISTATS 2025.](https://arxiv.org/abs/2412.16802)
>
> [6] [Choquette-Choo, C. A., Ganesh, A., Haque, S., Steinke, T., and Thakurta, A. Near exact privacy amplification for matrix mechanisms. ICLR 2025.](https://arxiv.org/abs/2410.06266)

---

> > ### Comment · Reviewer_k3aB · 2025-08-04
> >
> > I appreciate the authors responses. While the responses provided more evidence on the practical aspects of the study and slightly improved by judgement on this work, it is not enough for me to raise the score from 4 to 5. Thus, I decide to keep my original score.

---

### Official Review · Reviewer_ty3e · 2025-07-02

**Clarity:** 3
**Significance:** 3
**Originality:** 2
**Rating:** 4
**Confidence:** 3

**Summary:**

This paper studies differentially private training of 2-layer ReLU networks under the hidden state threat model by leveraging a convex approximation of the original non-convex objective. Two previously distinct ideas are integrated: convex approximation of 2-layer ReLU networks and privacy amplification by iteration. They propose a strongly convex surrogate model to facilitate the privacy analysis and apply NoisyCGD with disjoint mini-batches to achieve both strong privacy and utility guarantees.

**Questions:**

See weakness, especially,
could the authors focus on answering the following two questions.

1. Does network width affect the privacy and utility results in this setting? Is there some tradeoff between utility and network width $m$?

2. Could the author conclude the novelty of the paper?

**Ethical Concerns:**

["NO or VERY MINOR ethics concerns only"]

**Final Justification:**

I appreciate the authors responses. My main concern has been addressed. I will increase my score

**Quality:**

2

**Strengths And Weaknesses:**

Strength:

The topic of improving the theoretical performance of DP-SGD is both interesting and meaningful, particularly in the context of privacy-preserving machine learning. This paper contributes to this effort by providing utility analysis for 2-layer ReLU networks problem which is nonconvex.




Weakness:

1. The paper appears to be largely an integration and reapplication of results from previous studies. While the extension to ReLU networks is positioned as a novel contribution, the core theoretical results—such as the privacy bounds, convex reformulations, and the utility guarantees—are primarily based on established techniques and theorems from prior work. Although the context (applying these results to two-layer ReLU networks) may be new, the technical novelty and theoretical depth introduced by this paper appear limited.



2. The paper studies DP-SGD for 2-layer ReLU networks. While it confused me that even the studied nonconvex problem is equal to a convex approximation problem under some conditions (e.g., the networks width $m$ is large enough), to achieve DP, in each iteration, DP-SGD adds Gaussian noise to each weight, that is, in each iteration, there are $m$ noise added to the gradient since there is $m$ neurons and $m$ weights. Since there are $m$ neurons in the hidden layer, this results in adding $m$ independent noise terms to the gradient at each step. Intuitively, as $m$ increases, the amount of injected noise also increases, which should degrade the performance of DP-SGD. However, both the privacy and utility bounds presented in the paper appear to be independent of $m$. This  puzzling me a lot.

---

> ### Author Rebuttal · Authors · 2025-07-30
>
> We thank the reviewer for their positive summary and for describing our work as "interesting and meaningful."
>
> Regarding the trade-off between utility and model size: the reviewer is correct to point out that there is a trade-off between the utility and the parameter dimensionality in DP-SGD trained models. Notice that in our final approximation (loss function given in Eq. (3.6)), we have $P$ randomly chosen hyperplanes to approximate the ReLU network of width $m$. By clipping the per-example gradients of this loss, the sensitivity of the gradients is bounded, and the privacy analysis becomes independent of parameter dimensionality, as is common in privacy analysis of DP-SGD in general. For the utility analysis, the reviewer is correct to point out that the model size does not explicitly show up in the statement of Thm 4.1. That connection is given in the paragraph leading to the statement of Thm 4.1: with the choice of $P = O\big((n \log n\big)/d)$ random hyperplanes ($n$ denoting the number of data points and $d$ the feature dimension), we get the statement of Thm 4.1. We will add that assumption to the statement of Thm 4.1. Then, the resulting number of parameters is given by $p = P \cdot d$. Plugging that $p$ in the existing DP-SGD convergence bounds (Appendix E.1) gives the statement of Thm 4.1, so the parameter dimensionality is implicitly in there.
>
> Figures 4 and 5 in the appendix give an experimental illustration of the trade-off between the model size and the utility. While the model approximability seems to grow with growing $P$ (Fig. 4), the utility of the DP trained model does not seem to increase after a certain point (Fig. 5). The plot of Fig. 5 assumes an equal privacy budget and adjusts the DP noise accordingly.
>
> Regarding novelty: private learning of neural networks is important with the growing adoption of deep learning. With DP-SGD and its variants, the options are limited, and there is a big need for a hidden-state analysis of the privacy loss. Although we make use of existing works, we are the first to show that one can learn models with a utility similar to one hidden-layer ReLU networks in the hidden-state threat model of DP. In particular, we significantly improve upon the logistic regression that is the state-of-the-art model in this threat model of DP. We believe this is a very important first step towards new modes of privacy-preserving deep learning.
>
> The technical novelty comes from combining the theory from one field [1] (analysis of non-linear function learning) with the privacy analysis of iterative algorithms from another field [2]. There is a growing need for learning non-linear functions under the formal guarantees of DP, and we present an important milestone in that endeavour. While the reviewer might deem the combination trivial, the papers stem from 2018 and 2020, and no paper since has made this connection. We also remark that this also aligns with the NeurIPS 2025 guidelines on originality and novelty [3]. Unlike the prior work that focused on theory [1,2], we show that convexifying 2-layer ReLUs enables analysis in the hidden-state threat model of the DP setting where non-convex models fall short.
>
> [1] [Pilanci and Ergen. Neural networks are convex regularizers: Exact polynomial-time convex optimization formulations for two-layer networks. ICML 2020](https://arxiv.org/abs/2002.10553).
>
> [2] [Feldman, Mironov, Talwar, and Thakurta. Privacy amplification by iteration. Annual Symposium on Foundations of Computer Science (FOCS) 2018](https://arxiv.org/abs/1808.06651).
>
> [3] https://neurips.cc/Conferences/2025/ReviewerGuidelines

---

> > ### Comment · Reviewer_ty3e · 2025-08-07
> >
> > I appreciate the authors responses. My main concern has been addressed. I will increase my score

---

### Note · Authors · 2025-08-13

Thank you for taking the time to review our paper. We appreciate the positive remarks and are encouraged that two reviewers have increased their scores.

---

### Decision · Program_Chairs · 2025-09-17

**Decision:**

Accept (poster)

**Comment:**

This paper extends the hidden state privacy analyses only applicable currently in classification tasks for logistic regression models to 2-layer ReLU networks trained with DP stochastic gradient descent (DP-SGD). This is novel and important extension of private optimization analysis. Reviewers concerns on the DP noise increasing with the model size and novelty of the techniques have been addressed in the authors' rebuttal. However, a reviewer's concern about the practical significance of the proposed method still remains questionable. Overall, this is a solid paper with important contributions in the series of work in improving the analysis of DP optimization. This is the reason for my decision.